# Knowledge, attitudes, and fear of COVID-19 during the Rapid Rise Period in Bangladesh

**Mohammad Anwar Hossain[1], Md. Iqbal Kabir Jahid[1], K. M Amran Hossain[2]\*, Lori Maria Walton[3], Zakir Uddin[4], Md. Obaidul Haque[2‡], Md. Feroz Kabir[5‡], S. M. Yasir Arafat[6‡], Mohamed Sakel[7], Rafey Faruqui[8], Zahid Hossain[2‡]**

**1** Department of Microbiology, Jashore University of Science & Technology (JUST), Jashore, Bangladesh,
**2** Department of Physiotherapy, Bangladesh Health Professions Institute (BHPI), Dhaka, Bangladesh,
**3** Department of Physical Therapy, School of Health Sciences, University of Scranton, Scranton, Pennsylvania, United States of America, **4** School of Physical and Occupational Therapy, Faculty of Medicine, McGill University, Montréal, Canada, **5** Department of Physiotherapy & Rehabilitation, Jashore University of Science & Technology (JUST), Jashore, Bangladesh, **6** Department of Psychiatry, Enam Medical College Hospital, Dhaka, Bangladesh, **7** East Kent University NHS FT Hospitals, Canterbury, United Kingdom, **8** Kent & Medway NHS and Social Care Partnership Trust & University of Kent, Canterbury, United Kingdom

☯ These authors contributed equally to this work.
‡ These authors also contributed equally to this work.
\* amranphysio@gmail.com

**Data Availability Statement:** Data is available on www.kaggle.com/dataset/d4872de101ba5b232a347e2ff1fc0ba6f1483209b8c4c52d82aa96a863a8af71.

## Abstract

The study aims to determine the level of Knowledge, Attitude, and Practice (KAP) related to COVID-19 preventive health habits and perception of fear towards COVID-19 in subjects living in Bangladesh. Design: Prospective, cross-sectional survey of (n = 2157) male and female subjects, 13–88 years of age, living in Bangladesh. Methods: Ethical approval and trial registration were obtained before the commencement of the study. Subjects who volunteered to participate and signed the informed consent were enrolled in the study and completed the structured questionnaire on KAP and Fear of COVID-19 scale (FCV-19S). Results: Twenty-eight percent (28.69%) of subjects reported one or more COVID-19 symptoms, and 21.4% of subjects reported one or more co-morbidities. Knowledge scores were slightly higher in males (8.75± 1.58) than females (8.66± 1.70). Knowledge was significantly correlated with age (p < .005), an education level (p < .001), attitude (p < .001), and urban location (p < .001). Knowledge scores showed an inverse correlation with fear scores (p < .001). Eighty-three percent (83.7%) of subjects with COVID-19 symptoms reported wearing a mask in public, and 75.4% of subjects reported staying away from crowded places. Subjects with one or more symptoms reported higher fear compared to subjects without (18.73± 4.6; 18.45± 5.1). Conclusion: Bangladeshis reported a high prevalence of self-isolation, positive preventive health behaviors related to COVID-19, and moderate to high fear levels. Higher knowledge and Practice were found in males, higher education levels, older age, and urban location. Fear of COVID-19 was more prevalent in female and elderly subjects. A positive attitude was reported for the majority of subjects, reflecting the belief that COVID-19 was controllable and containable.

**Funding:** The author(s) received no specific funding for this work.

**Competing interests:** The authors have declared that no competing interests exist.

## Introduction

Bangladesh is among the top 20 countries in terms of confirmed cases of COVID-19, with a positive case rate of 19.09% - 22.91% as of June 1, 2020 [1]. However, questions remain regarding the actual number of cases and the scarcity of testing facilities [2]. There are also concerns about Bangladesh's ability to mount an effective response to the COVID-19 pandemic [3]. One newspaper also states [4] that Bangladesh is a developing economy and is mainly dependent on remittances, ready-made garments, and small trades. The country is mid-phase in a few financial megaprojects. Natural calamities and COVID-19 pose challenges for the Bangladeshi government and its residents at home and abroad [5]. Due to economic concerns, Bangladesh did not impose a countrywide lockdown. Instead, the authorities sub-sectioned the country into red, yellow, and green zones based on the level of community contamination [6]. Additionally, the government website for coronavirus briefing measures is being used to improve the situation, raising individual awareness by improving individual knowledge, attitudes, and practices, which has helped alleviate unnecessary fears and social stigmas [7].

Battling the COVID-19 pandemic is a lengthy process and requires the combined efforts of individuals and the government; adequate testing, isolation, and supportive treatment provision are the best ways to overcome the pandemic [8]. There is ongoing research to develop a vaccine. Nevertheless, measures to raise the general population's knowledge and implementation of recommended health practices are some of the best approaches to combating COVID-19 [9].

The World Health Organization (WHO) [10] stated that only 15% of cases were projected to have severe symptoms, and one-third of the severe cases required critical care; the main priority of the WHO is to mobilize resources to improve community healthcare practices. There is an emphasis on developing a community's receptiveness to staying at home. Moreover, the WHO raised concerns regarding mental health needs [11]. Mental health needs related to COVID-19 are emerging regardless of age, occupation, and education and are related to isolation, financial uncertainty, quarantine effects, excessive time spent online, gaming, physical inactivity, insomnia, anxiety, depression, and fear of COVID-19 [12]. The study also suggests that extreme fear and anxiety led individuals in China to have more physical and psychological signs, even with mild to no symptoms reported. Bangladesh reported a few cases of suicide due to extreme fear of COVID-19, with some cases showing negative outcomes after the administration of the real-time polymerase chain reaction (RT-PCR) test postmortem [13].

Bangladesh responded relatively early in March 2020, with no cases for nearly a week. The subsequent arrival of travelers from Italy who defied quarantine regulations could have been the source of the virus [14]. Besides, religious gatherings and the lack of travel restrictions are considered the primary reasons for the sharp upward projections in COVID-19 cases [15]. A population-based study was required to determine general knowledge about the disease and what practices were being taken by Bangladeshi individuals to combat COVID-19. Fear is thought to be one of the main contributors to mass anxiety and depression. Fear has been shown to predict inadequate health overall, insomnia, and the suppression of immunity. Other influencing factors of anxiety and depression include occupation, knowledge, attitudes, and practice of health-related habits, as well as other environmental indicators [12]. Determining knowledge, attitudes, practices (KAP), and fear will provide a glimpse of how Bangladesh is responding to the pandemic in this state of rising cases. This will further help to evaluate their overall preparedness. The COVID-19 crisis is assumed to be a long-term process, and the only way to battle the pandemic is to know the right information and practice the recommended health advisories. It is also necessary to examine the relationship among demographic variables

with KAP and fear to explore the in-depth understanding of factors contributing to the preparedness for and response towards COVID-19.

The study objectives were to determine the level of knowledge, attitudes, and practices related to COVID-19 preventive health habits and the underlying fear of COVID-19 in the Bangladeshi population and how they are affected by socio-demographic factors.

## Methodology

### Study design and participants

This study was a prospective cross-sectional survey conducted online through a structured questionnaire from April 4 to May 2, 2020. Both male and female Bangladeshi subjects, and aged 13 to 90 years, were able to respond to the questionnaire and were eligible for the study. Subjects with an intellectual disability or an inability to communicate were excluded from the survey.

### Questionnaire

A structured questionnaire has been designed by the authors to fulfill the objectives of the study. The questionnaire included socio-demographic variables (Table 1), questions on KAP, and fear. Questions related to KAP adapted from the survey questions used in a study conducted during a period of rapid case increases in China [16]. The KAP section of the questionnaire related to a total of 12 score knowledge questions on COVID-19, categorical answers to attitudes towards the control of the pandemic, and practices of wearing masks and avoiding public gatherings. The co-morbidities and symptoms of COVID-19 were obtained from WHO resources and asked to the respondents' whether present or not [17]. The Fear of COVID-19 Scale (FCV 19S) was used and reported to be valid and reliable in measuring fear attributed to coronavirus disease [18]. The questionnaire complied with the forward and back-translation into Bangla by a bilingual British researcher and sent to two renounced bilingual epidemiologists in Bangladesh to examine the difference and suitability of the questionnaire. Also, a pilot study was conducted before the commencement of the research.

### Ethical issues and trial registration

Ethical permission was obtained from the Institutional review board (BPA IPRR/IRB/29/03/2020/021) of the Institute of physiotherapy, rehabilitation, and research (IPRR). Participation was voluntary, consent was obtained, and confidentiality of the information was assured. The trial registration was obtained prospectively from a primary trial registry of the WHO (CTRI/2020/04/024413).

### Data collection procedure

From April to May 2020, the questionnaire was disseminated online, and through email and social media, targeting students, professionals, and public groups on Facebook. Recipients of the questionnaire were encouraged to complete it themselves and to send it to family members and neighbors for completion. A video tutorial was also provided to ensure an appropriate response. For illiterate family members, another member assisted them in responding to the questionnaire. The survey was requested to be sent back after completion. A total of 3500 questionnaires were sent, and 2200 questionnaires were returned. The data auditor found 2157 responses that could be included in the study and analyzed. The respondents were from all areas of Bangladesh and may represent the whole population.

Table 1. Relationship of demographic characteristics with Knowledge and fear.

| Characteristics | | Number of participants, n (%) | Knowledge mean± SD | t/Chi | p-value (2tailed) | Fear Score mean± SD | t/Chi | p-value (2tailed) |
|---|---|---|---|---|---|---|---|---|
| Gender | Male | 1166 (54.1%) | 8.75± 1.58 | 1.159 [a] | .247 | 18.07± 4.94 19.07± 5.04 | -4.647 [a] | .001** |
| | Female | 991 (45.9%) | 8.66± 1.70 | | | | | |
| Age | 13–20 years | 440 (20.4) | 8.59± 1.9 | 103.764 [b] | .071 | 18.45± 5.15 18.26± 4.99 18.74± 5.35 18.60± 4.79 18.89± 4.58 19.00± 4.89 19.44± 4.96 20.00± 4.35 | 192.155 [b] | .422 |
| | 21–30 years | 713 (33.1) | 8.81± 1.54 | | | | | |
| | 31–40 years | 355 (16.5) | 8.55± 1.74 | | | | | |
| | 41–50 years | 327 (15.2) | 8.80± 1.47 | | | | | |
| | 51–60 years | 226 (10.5) | 8.88± 1.29 | | | | | |
| | 61–70 years | 75 (3.5) | 8.33± 1.83 | | | | | |
| | 71–80 years | 18 (.8) | 8.44± 1.79 | | | | | |
| | 80–90 years | 3 (.1) | 8.67± 1.52 | | | | | |
| Education | Illiterate | 58 (2.7) | 8.83± 2.0 8.17± 2.13 8.54± 1.85 8.79± 1.63 8.86± 1.31 8.82 ± 1.21 | 140.943 [b] | .0001*** | 19.81± 3.69 18.68± 5.22 19.15± 5.25 18.51± 4.88 18.14± 4.84 17.99± 5.39 | 267.282 [b] | .001** |
| | primary education | 183 (8.5) | | | | | | |
| | Secondary school | 423 (19.6) | | | | | | |
| | Higher secondary | 696 (32.3) | | | | | | |
| | Graduation | 596 (27.6) | | | | | | |
| | Post-graduation | 201 (9.3) | | | | | | |
| Geography | Barisal | 203 (9.2) | 8.88± 1.16 9.17± 1.24 8.51± 2.09 8.90± 1.24 8.46± 1.62 8.56 ± 1.45 8.97± 1.68 8.46± 1.64 | 472.435 [b] | .0001*** | 18.75± 5.39 18.79± 4.37 19.15± 5.02 18.42± 4.50 18.96± 4.72 18.39± 4.75 17.10± 5.59 16.51± 6.28 | 1355.41 [b] | .0001*** |
| | Chittagong | 334 (15.2) | | | | | | |
| | Dhaka | 499 (23) | | | | | | |
| | Mymensingh | 103 (6) | | | | | | |
| | Khulna | 302 (14) | | | | | | |
| | Rajshahi | 408 (18.7) | | | | | | |
| | Rangpur | 207 (9.4) | | | | | | |
| | Sylhet | 101 (4.5) | | | | | | |
| Public service | Public Servant | 150 (7) | 8.93± 1.52 8.69± 1.65 | 1.738 [a] | .082 | 17.68± 4.97 18.60± 5.01 | -2.16 [a] | .031 * |
| | Non-public servant | 2007 (93) | | | | | | |
| Healthcare profession | Healthcare provider | 87 (4) | 8.99± 1.05 8.70± 1.65 | 1.623 [a] | .105 | 18.62± 4.36 18.53± 5.03 | .168 [a] | .867 |
| | Non-Healthcare | 2070 (96) | | | | | | |
| Business or work in a crowd | Yes | 294 (13.4) | 8.80± 1.46 8.70± 1.67 | .968 [a] | .333 | 17.91± 4.85 18.63± 5.03 | -2.29 [a] | .022 * |
| | No | 1863 (86.4) | | | | | | |
| Study | Students | 930 (43.1) | 8.83± 1.53 8.62± 1.71 | 2.905 [a] | .004** | 18.46± 5.11 18.59± 4.93 | -.607 [a] | .544 |
| | Non-students | 1227 (56.9) | | | | | | |
| Diagnosed COVID 19 in community | Yes | 166 (7.7) | 8.80± 2.02 8.70± 1.60 | .704 [a] | .093 | 18.29± 5.75 18.55± 4.94 | -.650 [a] | .516 |
| | No | 1991 (92.3) | | | | | | |
| COVID Symptoms (last 14 days) | Yes | 619 (28.7) | 9.04± 1.20 8.58± 1.78 | 5.9 [a] | .0001*** | 18.73± 4.6 18.45± 5.1 | 1.1.4 [a] | .002** |
| | No | 1536 (71.3) | | | | | | |

(Continued)

**Table 1.** (Continued)

| Characteristics | | Number of participants, n (%) | Knowledge mean± SD | t/Chi | p-value (2tailed) | Fear Score mean± SD | t/Chi | p-value (2tailed) |
|---|---|---|---|---|---|---|---|---|
| **Co-morbidity present** | Yes | 463 (21.4) | 9.03± 1.26 8.62± 1.72 | 4.8 [a] | .0001*** | 18.27± 4.8 18.61± 5.04 | -1.2 [a] | .229 |
| | No | 1692 (78.6) | | | | | | |

[a]Independent t test

[b] Chi-square test

* Significant with p < .05

** Significant with p < .005

*** Significant with p < .001

## Statistical analysis

Descriptive statistics were employed for correct answers to knowledge, and diverse attitudes and practices were presented. Knowledge, fear scores, attitudes, and practice variables of respondents were presented and compared with independent sample t-tests or chi-square tests to determine associations (Tables 1 and 3) between continuous data (knowledge and fear score) and categorical or nominal data (demographic variables) [19]. Binary logistic regression analysis using dichotomous demographic variables as dependent variables and knowledge and fear scores as covariates (Table 2) was performed to measure the relationship between categorical dependent variables and continuous variables. The chi-square test was employed to

**Table 2. Results of Binary logistic regression on factors associated with Knowledge and fear.**

| Variables | Knowledge | | | | Fear | | | |
|---|---|---|---|---|---|---|---|---|
| | Chi | Coefficient | OR | p | Chi | Coefficient | OR | p |
| **Gender Male vs Female** | 31.127 | -.163 | .850 | .002** | 81.573 | -.163 | .850 | .0001*** |
| **Age 13–40 years vs 41–88 years** | 14.544 | -.842 | .431 | .267 | 34.549 | -.843 | .430 | .151 |
| **Age 41–60 years vs 13–40 and 61–88 years** | 14.517 | 1.064 | 2.897 | .269 | 30.155 | 1.065 | 2.901 | .307 |
| **Age 61–88 years vs 13–60 years** | 16.484 | 3.066 | 21.448 | .170 | 36.788 | 3.067 | 21.469 | .099 |
| **Education Primary vs secondary to bachelor and above** | 37.070 | 2.072 | 7.942 | .0001*** | 47.470 | 2.073 | 7.950 | .009** |
| Education secondary and higher secondary vs primary, bachelor and above | 17.745 | -.075 | .928 | .124 | 75.208 | -.075 | .928 | .0001*** |
| **Education Bachelor and above vs primary to higher secondary** | 42.001 | .535 | 1.707 | .0001*** | 82.774 | .543 | 1.706 | .0001*** |
| **Geography Dhaka vs other 7 divisions** | 68.689 | 1.200 | 3.319 | .0001*** | 241.448 | 1.201 | 3.323 | .0001*** |
| **Public servant vs non-public servant** | 23.533 | 2.593 | 13.367 | .024* | 64.353 | 2.594 | 13.380 | .0001*** |
| **Health professionals vs non-health professionals** | 18.553 | 3.168 | 23.770 | .100 | 45.729 | 3.169 | 23.793 | .014* |
| **Business or work in a crowd vs all other professions** | 25.839 | 1.849 | 6.355 | .011* | 92.950 | 1.846 | 6.337 | .0001*** |
| **Students vs all other professions** | 23.841 | .277 | 1.320 | .021* | 54.226 | .277 | 1.319 | .001** |
| **Symptom present vs absent** | 229.083 | -1.355 | .258 | .0001*** | 217.997 | -1.356 | .258 | .0001*** |
| **Co-morbidity present vs absent** | 206.936 | -1.640 | .194 | .0001*** | 172.332 | -1.641 | .194 | .0001*** |
| **Believe in control agree vs disagree** | 102.858 | -.719 | .487 | .0001*** | 379.617 | -.720 | .487 | .0001*** |
| **Bangladesh can win agree vs disagree** | 95.688 | -.114 | .892 | .0001*** | 297.923 | -.114 | .892 | .0001*** |
| **Go to crowd yes vs no** | 16.816 | -5.726 | .003 | .157 | 20.129 | -5.727 | .003 | .825 |
| **Wear mask yes vs no** | 59.405 | -1.640 | .194 | .0001*** | 167.114 | -1.638 | .194 | .0001*** |

* Significant with p < .05

** Significant with p < .005

*** Significant with p < .001; vs means versus

**Table 3. Relationship among attitude and practice with demographic variables.**

| Variables | | Attitude | | | | | | Practice | | | | | |
|---|---|---|---|---|---|---|---|---|---|---|---|---|---|
| | | A1: Belief in Control | | | A2: Bangladesh can win | | | P1: Go to crowd | | | P2: Wear Mask | | |
| | | Agree | Disagree | Chi value | Yes | No | Chi | Yes | No | Chi value | Yes | No | Chi |
| **Gender** | Male Female | 751 (34.8) 586 (27.2) | 415 (19.2) 405 (18.8) | 6.33* | 487 (22.6) 394 (18.3) | 679 (31.5) 597 (27.7) | .895 | 346 (16.0) 178 (8.3) | 818 (37.9) 808 (37.5) | 48.28*** | 1012 (46.9) 794 (36.8) | 154 (7.1) 197 (9.1) | 17.5*** |
| **Age** | 13–40 years 41–60 years 61–90 years | 925 (42.9) 357 (16.6) 55 (2.5) | 583 (27.0) 196 (9.1) 41 (1.9) | 2.71 | 595 (27.6) 250 (11.6) 36 (1.7) | 913 (42.3) 303 (14.0) 60 (2.8) | 6.01 | 350 (16.2) 153 (7.1) 21 (1.0) | 1158 (53.7) 400 (18.5) 75 (3.4) | 8.04 | 1255 (58.2) 474 (22.0) 77 (3.6) | 253 (11.7) 79 (3.7) 19 (.9) | 2.76 |
| **Education** | Primary SSC & HSC[1] Bachelor ≤[2] | 125 (5.8) 677 (31.4) 535 (24.8) | 116 (5.4) 442 (20.5) 262 (12.1) | 20.46*** | 63 (2.9) 452 (21.0) 366 (17.0) | 178 (8.3) 667 (30.9) 431 (20.0) | 30.16*** | 63 (2.9) 266 (12.3) 195 (9.0) | 178 (8.3) 853 (39.5) 602 (27.9) | 16.78** | 201 (9.3) 929 (43.1) 676 (31.3) | 40 (1.9) 190 (8.8) 121 (5.6) | 1.13 |
| **Geography** | Dhaka Other parts | 264 (12.2) 1073 (49.7) | 235 (10.9) 585 (27.1) | 22.71*** | 168 (7.8) 713 (33.1) | 331 (15.3) 945 (43.8) | 13.84*** | 69 (3.2) 455 (21.1) | 430 (19.9) 1203 (55.7) | 41.41*** | 397 (18.4) 1409 (65.3) | 102 (4.7) 249 (11.5) | 8.28** |
| **Public Servant** | | 108 (5.0) | 42 (1.9) | 6.86** | 82 (3.8) | 68 (3.2) | 12.75*** | 43 (2.0) | 107 (5.0) | 2.14 | 130 (6.0) | 20 (.9) | 1.02 |
| **Healthcare provider** | | 57 (2.6) | 30 (1.4) | .480 | 40 (1.9) | 47 (2.2) | .989 | 28 (1.3) | 59 (2.7) | 3.31 | 79 (3.7) | 8 (.4) | 3.33 |
| **Work in a crowd** | | 208 (9.6) | 86 (4.0) | 11.09** | 113 (5.2) | 181 (8.4) | .817 | 123 (5.7) | 171 (7.9) | 57.62*** | 272 (12.6) | 22 (1) | 19.31*** |
| **Students** | | 560 (26.0) | 370 (17.2) | 2.172 | 357 (16.6) | 573 (26.6) | 4.08* | 196 (9.1) | 729 (33.8) | 11.22** | 755 (35.0) | 175 (8.1) | 7.77** |
| **COVID 19 in community** | | 144 (5.3) | 52 (2.4) | 3.416 | 45 (2.1) | 121 (5.6) | 14.04*** | 55 (2.5) | 110 (5.1) | 8.196* | 140 (6.5) | 26 (1.2) | .049 |
| **Symptoms present** | | 442 (20) | 177 (8.2) | 54.26*** | 248 (11.4) | 371 (17.1) | .028 | 160 (7.4) | 459 (21.2) | 15.08 | 533 (24.7) | 86 (3.9) | 2.97* |
| **Co-morbidity present** | | 350 (16.2) | 113 (5.2) | 39.22*** | 188 (8.7) | 275 (12.7) | .221 | 137 (6.3) | 326 (15.1) | 27.74** | 396 (18.3) | 67 (3.1) | 9.97 |
| **Total Knowledge Score** | | 8.9± 1.3 | 8.3± 2 | 86.09*** | 8.9± 1.2 | 8.5± 1.8 | 67.78*** | 8.6± 1.5 | 8.7± 1.6 | 98.671 | 8.7± 1.6 | 8.5± 1.7 | 62.28 |
| **Total Fear Score** | | 17.9± 4.9 | 19.4± 5.0 | 296.79*** | 18.2± 4.8 | 18.7± 5.1 | 135.71* | 17.7± 5.4 | 18.7± 4.8 | 375.38*** | 18.5± 4.7 | 18.2± 6.3 | 186.37*** |

* Significant with p < .05

** Significant with p < .005

*** Significant with p < .001

[1] Secondary and Higher Secondary school certificate

[2] equal and above

determine the relationship between attitude and practice with demographic variables, and knowledge and fear score (Table 3). Data analysis was completed using the IBM Statistical package for the social sciences (SPSS) version 20.0. The alpha level of significance was set at P < .05.

## Results

### Socio-demographics

Among 2157 respondents, 1166 (54.1%) were male and 991 (45.9%) were female. The mean population age was 33.48±14.65 years. The participants' ages ranged from 13 years to 88 years, and the majority of the respondents were aged 21–30 years (33.1%). Respondents were categorized as adolescents (10–20 years), youth (21–40 years), adults (41–60 years), and elderly

(above 60 years). There was a larger response among those with higher secondary education (32.3%) and graduates (27.6%). A total of 11.2% of respondents were either undergoing primary education or reported low levels of literacy. Respondents were from all divisions of Bangladesh; the highest response was from Dhaka (23%), and the lowest was from Sylhet (4.5%). The majority of the respondents were non-public servants (93%), 4% were healthcare professionals, 13.4% worked or did business in a crowded place, 43.1% were students, and 7.7% of the respondents reported that a relative, colleagues or a neighbor had been diagnosed with COVID-19. Other sociodemographic profiles are described in Table 1.

## COVID-19 symptoms and co-morbidity

Multiple response analyses found that 28.69% of the respondents (n = 619) reported one or more symptoms related to COVID-19 in the last 14 days, but none reported completing a COVID-19 test during the response. The most prevalent symptoms were dry cough 19.5% (n = 121), cough with sputum (14.2%), sore throat (10%), fever of more than 100˚ F (4.7%), anosmia or taste loss (4.5%), and shortness of breath (3.7%); 4 patients were diagnosed with pneumonia, and 3 patients were hospitalized for pneumonia. Multiple response analyses also found 463 respondents (21.4%) who reported one or more co-morbidities, including diabetes (7.1%), chronic obstructive pulmonary disease (COPD) (1.9%), and heart disease or hypertension (2.8%). Nine subjects reported a chronic neurological disability, including stroke; 20 subjects reported chronic kidney disease (CKD), and 4.6% reported chronic smoking habits.

## Knowledge

In the population knowledge score, the mean was 8.71 out of 12, and the standard deviation was 1.64. Knowledge regarding COVID was similar in both males (8.75± 1.58) and females (8.66± 1.70). There was a significant relationship found between knowledge scores and age (p < .005), an education level (p < .001), and geographical distribution (p < .001). No significant differences in knowledge scores were found in the following comparisons: between public servants (8.93± 1.52) and others (8.69± 1.65); between healthcare professionals (8.99± 1.05) and others (8.70± 1.65); working in a crowd (8.80± 1.46) or working alone (8.70± 1.67); or in people who reported COVID-19-positive relatives, friends or colleagues (8.80± 2.02) vs. those with associates without COVID-19 (8.70± 1.60). Significant differences were found between subjects with symptoms of COVID-19 (9.04± 1.20) and subjects without COVID-19 symptoms (8.58± 1.78) (p < .001). Additionally, a significant difference (p < .001) was found in knowledge scores between subjects with co-morbidities (9.03± 1.26) and subjects without co-morbidities (8.62± 1.72). The detailed associations are available in Table 1.

Binary logistic regression analysis showed a significant correlation between knowledge scores and gender (p < .005). Logistic regression associations were found between knowledge and education levels, with the lowest knowledge scores found in primary education compared to all other education groups (p < .001). Dhaka "urban dwellers" reported significantly higher knowledge of COVID-19 symptoms and precautions than did subjects from rural areas of Bangladesh (p < .001). Knowledge and education levels were directly associated, with Bachelor of Science (BSc) degree holders reporting higher knowledge of COVID-19 symptoms and precautions than any other education group (p < .005). Public servants reported higher knowledge than did other non-public servant groups (p < .05), and students reported higher knowledge of COVID-19 symptoms and precautions than did other non-student groups (p < .05). Subjects without symptoms showed a significant inverse relationship with knowledge than did those with symptoms (p < .001) (Table 2).

## Attitudes

Attitudes were measured concerning "beliefs" regarding whether Bangladesh can overcome the challenge of COVID-19 or "positive synergy" towards disease control. Females reported a higher belief that COVID-19 could be controlled (p < .05). Similarly, graduates or more qualified respondents were confident that COVID-19 can be controlled (p < .001) and agreed that Bangladesh was capable of overcoming the challenge (p < .001). The majority of subjects who identified as public servants in Bangladesh also reported belief that the disease was controllable (p < .05); however, they did not believe COVID-19 would be overcome easily (p < .001). The subjects with a higher knowledge score of COVID-19 and a higher score on the COVID-19 Fear Scale also showed higher scores in "belief" that the virus was controllable (p < .005) and that eradication of the virus nationwide would be achieved (p < .001). Subjects with COVID-19 symptoms and co-morbidities reported a higher prevalence of the "belief" that the virus was both controllable and containable (p < .001). Details are presented in Table 3. Additionally, Table 2 shows that the subjects with a higher knowledge score of COVID-19 and a higher score on the COVID-19 Fear Scale also showed higher scores on the "belief" that the virus was controllable (p < .001) and that eradication of the virus nationwide would be achieved (p < .001).

## Practices

Practices were measured by the report of the subject's attendance in crowded areas and reports of wearing a mask. The majority of female subjects in the study followed the practice of staying home (37.5%) and wearing a mask (36.8%) to prevent the spread of COVID-19 (p < .001). Similarly, 27.9% of qualified personnel who were graduates and above reported staying home and avoiding crowded spaces (p < .005). The majority of the population from Dhaka followed the health advisory by staying home (55.7%) (p < .001) and reported wearing a mask (65.3%) (p < .005). No significant relationship was found between knowledge score and practice, but a highly significant association was found between fear scores and adhering to the health advisory and between fear scores and reporting mask-wearing (p < .001). The majority of subjects with COVID-19 symptoms reported wearing a mask (p < .05) but also reported going to a crowded place. The majority of subjects with co-morbidities also reported staying at home but did not report wearing a mask (p < .005) (Table 3). Table 2 explores a significant association (p < .001) between knowledge scores and wearing masks.

## Fear

The population means the fear score was 18.53 out of 35, with a standard deviation of 5.013. Fear scores were strongly associated with gender, education and geography (p < .001), with females reporting a higher score (19.07± 5.04) and respondents aged 61–70 years, 71–80 years and 80–90 years reporting a higher score of fear of contracting COVID-19 (19.00± 4.89; 19.44± 4.96; 20.00± 4.35). Dhaka urban dwellers also reported a higher fear status than did rural dwellers (19.15± 5.02) (p < .001). The demographic relationship of fear scores is listed in Table 1. Binary logistic regression found gender differences in fear scores (p < .001). Other regressions are described in Table 2. An indirect, strong, but significant relationship (p < .001) was found between the fear scores and practices of recommended health advisory habits of subjects (Table 3). There were significant differences (p < .005) in fear scores between subjects with symptoms and those without symptoms (18.73± 4.6; 18.45± 5.1) (Table 1). Inverse relationships were found among persons with positive COVID-19 symptoms and fear scores (p < .005). Table 2 shows a significant association (p < .001) between fear scores and wearing masks.

## Discussion

The study intended to explore the knowledge, attitudes, and practices of recommended health advice for the prevention of COVID-19 and to explore the impact of fear towards contracting COVID-19 on people living in Bangladesh. There is little to no research published on this important topic for Bangladesh to date. The study covered every geographical area of Bangladesh according to administrative distribution and provided a glimpse of the time frame.

Questions of knowledge, attitudes, and practices have been used following a Chinese study [16], which was relevant in terms of geographical distribution, as both Bangladesh and China are on the Asian continent. Additionally, the time was relevant, the questionnaire was prepared for a rapid rise in cases, and Bangladesh was the process of experiencing a rapid rise in cases from April to May 2020 as per WHO case definition [17]. The questionnaire development and translation completed in a structured and standard process. The fear scale was a valid questionnaire [18] and until the start of data collection, the Bangla language validation had not been published. Therefore, all the questionnaires have been translated with the WHO guidelines for translating questionnaires [20].

Among the respondents, there were more males (54.1%) than females (45.9%). The responses per age group were distributed as follows: 20.4% among those aged 13–20 years, 33.1% aged 21–30 years, 16.5% aged 31–40 years, 15.2% aged 41–50 years, 10.5% aged 51–60 years, 3.5% aged 61–70 years, .8% (n = 18) aged 71–80 years and .1% (n = 3) aged 80–90 years. Fifty-eight respondents (2.7%) were illiterate, 183 persons (8.5%) had only primary education, 423 respondents (19.6%) had secondary education, 696 persons (32.3%) had a higher secondary degree, 596 respondents (27.6) obtained a graduate degree, and 201 persons(9.3%) had postgraduate degrees. The highest response of COVID-19 cases was from Dhaka (23%). The Institute for Epidemiology, Disease Control, and Research (IEDCR) COVID-19 update reported [21] that Dhaka had the majority of SARS-CoV-2 cases, and it is considered to have the most challenges regarding the level of practice of healthcare advisory precautions [22]. The baseline characteristics of subjects in comparative studies varied across the world. Studies in India, China, and Egypt had more responses from females, and the USA had more responses from males [16, 22–25] Indian, Chinese, and Egyptian studies had similar responses by age group and education, while the USA study reported a mean age higher than that in our study. Our study found a satisfactory level of knowledge by gender, geography, occupation, and education (Table 1) and relatively higher fear scores than those observed in similar studies across the world. One study in China showed similar scores of fear by age concerning knowledge and occupation, while another study completed in India reported that 80% of people in need of mental health care for COVID-19 experienced fear, anxiety, and depression [16, 25].

Twenty-nine percent (28.69%) of the respondents (n = 619) reported one or more symptoms related to COVID-19 in the last 14 days, including cough 19.5% (n = 121), cough with sputum (14.2%), sore throat, (10%), fever (4.7%), anosmia or taste loss (4.5%), and shortness of breath (3.7%). The symptoms were related to COVID-19, as per the CDC [26]. The WHO South-East Asia region reported the test positivity in Bangladesh to be 20%, with positive tests being reported only for a person with one or more COVID-19-related symptoms [2]. Eleven percent (11%) of subjects reported co-morbidities, including subjective disabilities. Besides, 4.4% of the respondents were over 60 years of age. The WHO South-East Asia region country profile and the IEDCR COVID-19 update states that the number of deaths is higher among elderly persons, males, and those with pre-existing co-morbidities in Bangladesh [21, 27]. Overall, we found a low number of elderly patients with symptoms, low reported levels of co-morbidities, and a slightly higher rate of infection among males than among females. The reason behind the higher rate of infection among males is their greater exposure outside. At that

time, Bangladesh was in a state of "movement restriction", and no nationwide "lockdown" had been imposed [6, 15], so more male cases were expected. The symptoms were matched with the WHO statements [17] of symptoms until then, and many cases were found to be relevant, but they were not tested. Many reports have been published on unwillingness, droughts, and fear regarding COVID-19 testing in Bangladesh [3, 4].

Knowledge regarding COVID-19 by the subject was satisfactory and similar across age, gender, and occupation. There were a few variations in perspectives by occupation. Young, graduates and urban dwellers had more knowledge than did older adults, those with lower education, and those living in rural areas. Several similar articles in the preprint found that more than half of the respondents reported "good knowledge" of COVID-19, with age and education showing a significant linear association with knowledge [27–29]. This study is similar to one study in China that found a significant relationship between knowledge and age and knowledge and educational level, with males reporting higher levels of knowledge than females regarding COVID-19 symptoms, precautions, and health advisory practices [16]. However, in our study, subjects living in Bangladesh reported similar knowledge for both males and females regarding COVID-19 symptoms, precautions, and health advisory practices. This finding may be attributed to a similar degree of access to information through print and electronic media and internet access in Bangladeshi populations, as the country's digital gateway is currently being prioritized.

Overall, a high prevalence of "positive attitudes" among the subjects regarding disease control was reported. Female subjects and subjects with higher levels of education were more likely to believe that COVID-19 can be controlled, but they doubted the ability of Bangladesh to contain it. Subjects with "good knowledge" or "high scores for fear" were more likely to believe that COVID-19 can be controlled and that a collective effort can contain the spread of the disease. Similar studies in Bangladesh, India, and China all found similar results regarding the relationship between knowledge and fear of COVID-19 regarding "practices", [16, 25, 30], and our study reported similarities to previous studies across the world [23, 24]. Our study found that 37.5% of women reported "staying home", and 36.8% reported wearing masks in public places. The majority of the population outside of Dhaka, i.e., those living in the more rural regions, reported staying home (55.7%) and wearing a mask (65.3%). However, no statistical relationship was found between knowledge scores and practices. This is similar to results reported in studies in both India and China. However, our study did find subjects with high fear scores and who were also more likely to follow good preventive practices, as recommended by the health advisory.

The fear score was significantly associated with female gender, higher education, and urban dwelling. Senior citizens aged 61–70 years, 71–80 years, and 80–90 years reported the highest fear scores among individuals in all categories. One study suggests that fear comes from a longer duration of isolation, greater movement restriction, and greater reactivity to news and rumors from social media [31]. Women, senior citizens, and young adults had limited movement, were isolated from quarantining, and were attached to media. A Study reports that fear and stress can lead to insomnia and psychological illness [25]. Fear is an important component in both positive and negative ways, as illustrated in the positivity explained earlier; hence, there have been cases of suicide due to the fear of COVID-19 in Bangladesh [13].

However, although important as an indicator, this was not evaluated in this study and may be considered a limitation of the study. Our study faced challenges regarding structured questionnaires, reporting, and resources. Limitations included the response rate (62.85%) and the completion of the questionnaire by none COVID-19-positive individuals. We recommend that future studies include information on the long-term observation of corvid-19-positive

cases or cases with symptoms with respect to movement, function, physical signs, mental health, and quality-of-life issues.

## Conclusion

In a resource-challenged country such as Bangladesh, individual knowledge, positive attitudes, and practices of suggested precautionary and preventive health advisories are crucial to controlling the vicious community transmission of COVID-19. The study found that knowledge levels were adequate in the majority of the population and were directly and significantly related to higher education levels, younger age, and female gender. There were positive attitudes among respondents regarding the control of the disease and the overcoming of challenges related to COVID-19 in Bangladesh. The majority of the population had high fear scores, with significantly higher scores found in women and elderly adults. Surprisingly, those with higher fear scores had good practices of staying at home and wearing masks. Future studies on explanatory issues related to activity, function, social issues, and quality of life might add more insight into the bio-psychological impact of COVID-19 in the most densely populated country in the world.

## Supporting information

**S1 File. English questionnaire.**
(DOCX)

**S2 File. Bangla questionnaire.**
(DOCX)

## Acknowledgments

Authors acknowledge Rubayet Shafin and Ahnaf Al Mukit, research assistants for their contribution in data collection and input, also students of Bangladesh health professions institute helped in collecting data. The authors are also grateful to Professor Dr. Md. Forhad Hossain for his support in statistical analysis.

## Author Contributions

**Conceptualization:** Mohammad Anwar Hossain, Md. Iqbal Kabir Jahid, K. M Amran Hossain, Lori Maria Walton, Zakir Uddin, Md. Obaidul Haque, Md. Feroz Kabir, S. M. Yasir Arafat, Mohamed Sakel, Rafey Faruqui, Zahid Hossain.

**Data curation:** Mohammad Anwar Hossain, Md. Iqbal Kabir Jahid, K. M Amran Hossain, Zakir Uddin, Md. Obaidul Haque, Md. Feroz Kabir, Rafey Faruqui, Zahid Hossain.

**Formal analysis:** Mohammad Anwar Hossain, Md. Iqbal Kabir Jahid, K. M Amran Hossain, Lori Maria Walton, Md. Feroz Kabir, S. M. Yasir Arafat, Mohamed Sakel, Rafey Faruqui, Zahid Hossain.

**Funding acquisition:** Mohammad Anwar Hossain, K. M Amran Hossain, Md. Feroz Kabir, Rafey Faruqui, Zahid Hossain.

**Investigation:** Md. Iqbal Kabir Jahid, K. M Amran Hossain, Lori Maria Walton, Zakir Uddin, S. M. Yasir Arafat, Mohamed Sakel.

**Methodology:** Mohammad Anwar Hossain, Md. Iqbal Kabir Jahid, K. M Amran Hossain, Lori Maria Walton, Zakir Uddin, Md. Obaidul Haque, Md. Feroz Kabir, S. M. Yasir Arafat, Mohamed Sakel, Rafey Faruqui, Zahid Hossain.

**Project administration:** Mohammad Anwar Hossain, K. M Amran Hossain, Md. Obaidul Haque, Mohamed Sakel.

**Resources:** K. M Amran Hossain, Lori Maria Walton, Zakir Uddin, Md. Obaidul Haque, Md. Feroz Kabir, Mohamed Sakel, Rafey Faruqui, Zahid Hossain.

**Software:** K. M Amran Hossain, Zahid Hossain.

**Supervision:** Mohammad Anwar Hossain, Md. Iqbal Kabir Jahid, K. M Amran Hossain, Lori Maria Walton, S. M. Yasir Arafat, Rafey Faruqui.

**Validation:** Mohammad Anwar Hossain, Md. Iqbal Kabir Jahid, Md. Obaidul Haque, Md. Feroz Kabir, Mohamed Sakel, Rafey Faruqui, Zahid Hossain.

**Visualization:** Mohammad Anwar Hossain, K. M Amran Hossain, Lori Maria Walton, Zakir Uddin, Md. Obaidul Haque, Mohamed Sakel, Zahid Hossain.

**Writing – original draft:** Mohammad Anwar Hossain, K. M Amran Hossain, Lori Maria Walton, Zakir Uddin, Md. Obaidul Haque, S. M. Yasir Arafat.

**Writing – review & editing:** Mohammad Anwar Hossain, Md. Iqbal Kabir Jahid, K. M Amran Hossain, Lori Maria Walton, Zakir Uddin, Md. Obaidul Haque, Md. Feroz Kabir, S. M. Yasir Arafat, Mohamed Sakel, Rafey Faruqui, Zahid Hossain.

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
