## [Decision Letter · Decision Letter 0]

27 Jul 2020

PONE-D-20-20624

Knowledge, Attitudes, and Fear of COVID-19 during the Rapid Rise Period in Bangladesh

PLOS ONE

Dear Dr. Hossain,

Thank you for submitting your manuscript to PLOS ONE. After careful consideration, we feel that it has merit but does not fully meet PLOS ONE’s publication criteria as it currently stands. Therefore, we invite you to submit a revised version of the manuscript that addresses the points raised during the review process.

We look forward to receiving your revised manuscript.

Kind regards,

Amir H. Pakpour, Ph.D.

Academic Editor

PLOS ONE

Journal Requirements:

3. Please include additional information regarding the survey or questionnaire used in the study and ensure that you have provided sufficient details that others could replicate the analyses. For instance, if you developed a questionnaire as part of this study and it is not under a copyright more restrictive than CC-BY, please include a copy, in both the original language and English, as Supporting Information. Moreover, please include more details on how the questionnaire was pre-tested, and whether it was validated.

4. In your Methods section, please provide additional information about the participant recruitment method and the demographic details of your participants. Please ensure you have provided sufficient details to replicate the analyses such as: a) the recruitment date range (month and year), b) a description of any inclusion/exclusion criteria that were applied to participant recruitment, c) a table of relevant demographic details, d) a statement as to whether your sample can be considered representative of a larger population, e) a description of how participants were recruited, and f) descriptions of where participants were recruited and where the research took place.

5. We noted in your submission details that a portion of your manuscript may have been presented or published elsewhere.

"The article is in the MedRxiv pre-print server https://www.medrxiv.org/content/10.1101/2020.06.17.20133611v1

" ext-link-type="uri" xlink:type="simple">https://doi.org/10.1101/2020.06.17.20133611"

Please clarify whether this publication was peer-reviewed and formally published. If this work was previously peer-reviewed and published, in the cover letter please provide the reason that this work does not constitute dual publication and should be included in the current manuscript.

6. Your ethics statement must appear in the Methods section of your manuscript. If your ethics statement is written in any section besides the Methods, please move it to the Methods section and delete it from any other section. Please also ensure that your ethics statement is included in your manuscript, as the ethics section of your online submission will not be published alongside your manuscript.

Reviewers' comments:

Reviewer's Responses to Questions

**Comments to the Author**

1. Is the manuscript technically sound, and do the data support the conclusions?

Reviewer #1: Partly

2. Has the statistical analysis been performed appropriately and rigorously? 

Reviewer #1: No

3. Have the authors made all data underlying the findings in their manuscript fully available?

Reviewer #1: Yes

4. Is the manuscript presented in an intelligible fashion and written in standard English?

Reviewer #1: No

5. Review Comments to the Author

Reviewer #1: The study entitled “Knowledge, Attitudes, and Fear of COVID-19 during the Rapid Rise Period in Bangladesh” has the strength of a large sample size (n=2157). However, the manuscript suffers from many flaws and I am not sure whether the authors can satisfactorily revise their work.

1. The authors have wrong statistical knowledge in the significance level. We do not use p=0.05, we use p0.05. That is, if the p-value is right at the 0.05, the comparison is not significant. However, the authors use = throughout the manuscript.

2. The authors conducted some ANOVA tests and did not further conduct the post-hoc analysis.

3. The authors used “r” to report the regression findings. However, “r” refers to Pearson’s’ r (or some use an r to represent Spearman’s rho). The r should not be used for regression models and the presentation is confusing and misleading.

4. The authors sometimes report a p-value at 0.000. However, it is impossible for a p-value to be exactly at 0.000. p0.001 should be used instead.

5. The information of questions on knowledge, attitude, and practice is vague and insufficient for me to evaluate. Specifically, the psychometric properties of these questions are unknown and how did the authors ensure that these questions developed from a Chinese study can be applicable to the present study?

6. The abbreviation of the Fear of COVID-19 Scale should be FCV-19S, not FCS. Also, the psychometric properties paper on Bangla version of the FCV-19S has been published. The authors should acknowledge this and cite properly.

7. The Introduction does not prepare the readers why the authors want to examine knowledge, attitude, practice, and fear for different subgroups using the demographics. Also, it is unclear why the authors wanted to examine the associations between these factors.

8. Line 207, I cannot understand why the people aged between 41 and 60 years can be defined as “young adults”. They are already middle age. Also, what is the meaning of “young” here? Do the authors want to say youth?

9. The authors reported the COVID 19 Symptoms and Comorbidity. However, the authors did not mention how they assessed these symptoms and comorbidity.

10. The authors mentioned that they used chi-square test in the Data analysis section; however, there are no results on chi-square values in the Results section.

11. The presentation of Table 2 is confusing. What are the “others” in the variables column? Also, the authors should explicitly indicate what the reference group is.

12. The authors mentioned that they performed binary logistic regression in the Data analysis section; however, I see no results on the logistic regression in the Results section.

13. The meanings of SSC and HSC in the Table 3 are unclear.

14. The Discussion reports redundant information; that is, the author repeated their results in the Discussion again. However, I did not see the authors “explain” their findings in the Discussion. They simply repeated the Results and then mentioned other similar studies’ findings.

15. There are some awkward sentences hinder the reading. For example, “Limitations response rates, correlation with fear and psychological issues, and completion of the questionnaire for COVID-19 positive. We recommend future studies to include information on limitation of movement, isolation and insomnia as it relates to psychological illness, as well as information regarding neurological signs and symptoms of patients and relationship to cognition and fear.”

16. The authors mentioned the age range of 13-90 in the Abstract. However, the age range is 13-88 in the Results section.

6. PLOS authors have the option to publish the peer review history of their article (what does this mean?). If published, this will include your full peer review and any attached files.

Reviewer #1: No

---

## [Author Response · Author response to Decision Letter 0]

6 Sep 2020

Dear Reviewers

Thank you for your comments. We appreciate your evaluation of our work and the manuscript. Please find our responses.

Reviewer’s comment: The study entitled “Knowledge, Attitudes, and Fear of COVID-19 during the Rapid Rise Period in Bangladesh” has the strength of a large sample size (n=2157). However, the manuscript suffers from many flaws and I am not sure whether the authors can satisfactorily revise their work

Author’s response: Thank you for your positive comments; we have revised the manuscript substantially.

Reviewers comment 1: 

1. The authors have wrong statistical knowledge in the significance level. We do not use p=0.05, we use p0.05. That is, if the p-value is right at the 0.05, the comparison is not significant. However, the authors use <=throughout the manuscript

Author response: We agree, we put p=<.05 in the manuscript to reflect “p is equal to less than point zero five”, we didn’t put it after the less symbol, but it is creating confusion, so we corrected to p<.05 

Reviewers comment 2: 

2. The authors conducted some ANOVA tests and did not further conduct the post-hoc analysis

Author response: Agreed, we excluded the ANOVA to examine the association between categorical and numerical data and placed with a “Chi-square test” as per statistician’s suggestion. https://pubmed.ncbi.nlm.nih.gov/23894860/

Reviewers comment 3: 

3. The authors used “r” to report the regression findings. However, “r” refers to Pearson’s’ r (or some use an r to represent Spearman’s rho). The r should not be used for regression models and the presentation is confusing and misleading.

Author response: We agreed and have made that change.

Reviewers comment 4: 

4. The authors sometimes report a p-value at 0.000. However, it is impossible for a p-value to be exactly at 0.000. p0.001 should be used instead

Author response: Agreed, in some cases, the value is .00001, so we turned to .000, but accepted the suggestion and changed to .0001 or .001

Reviewers comment 5: 

5. The information of questions on knowledge, attitude, and practice is vague and insufficient for me to evaluate. Specifically, the psychometric properties of these questions are unknown and how did the authors ensure that these questions developed from a Chinese study can be applicable to the present study?

Author response: Questions related to KAP adapted from the survey questions used in a study conducted during a period of rapid case increases in China (https://www.ncbi.nlm.nih.gov/pmc/articles/PMC7098034/). The KAP section of the questionnaire related to a total of 12 score knowledge questions on COVID-19, categorical answers to attitudes towards the control of the pandemic, and practices of wearing masks and avoiding public gatherings. The questionnaire had a wider acceptance and was retrieved by a couple of good quality papers in the same geographic area. Some papers used the same questionnaire in Malaysia (https://journals.plos.org/plosone/article?id=10.1371/journal.pone.0233668 ), in Egypt (https://link.springer.com/content/pdf/10.1007/s10900-020-00827-7.pdf), and in Saudi Arabia (https://www.ncbi.nlm.nih.gov/pmc/articles/PMC7266869/).

The questionnaire complied with the forward and back-translation into Bangla by a bilingual British researcher and sent to two renounced bilingual epidemiologists in Bangladesh to examine the difference and suitability of the questionnaire. Also, a pilot study was conducted before the commencement of the research. This information we added in the method section for better clarification. 

Moreover, the Fear of COVID-19 Scale (FCV 19S) was used and the Bangla versionreportedvalid and reliable tool with robust psychometric properties (see the link, https://link.springer.com/article/10.1007/s11469-020-00270-8). The psychometric properties of the Bangla version KAP not yet published. However, as mentioned earlier, the forward and back-translation into Bangla as well as a pilot study justified the use of KAP.

Also, the questions were the best fit for our geography, study objectives, and similarity of rapid raise situation. 

Reviewers comment 6: 

6. The abbreviation of the Fear of COVID-19 Scale should be FCV-19S, not FCS. Also, the psychometric properties paper on Bangla version of the FCV-19S has been published. The authors should acknowledge this and cite properly

Author response: Agreed and revised. We couldn’t use the Bangla version with psychometric validated question because it has been published in May 2020 https://www.ncbi.nlm.nih.gov/pmc/articles/PMC7213549/

And we started data collection in April 2020.

Reviewers comment 7: 

7. The Introduction does not prepare the readers why the authors want to examine knowledge, attitude, practice, and fear for different subgroups using the demographics. Also, it is unclear why the authors wanted to examine the associations between these factors.

Author response: Agreed, and revised in Page 4, as 

Determining knowledge, attitudes, practices (KAP),and fear will provide a glimpse of how Bangladesh is responding to the pandemic in this state of rising cases. This will further help to evaluate their overall preparedness. The COVID-19 crisis is assumed to be a long-term process, and the only way to battle the pandemic is to know the right information and practice the recommended health advisories. It is also necessary to examine the relationship among demographic variables with KAP and fear to explore the in-depth understanding of factors contributing to the preparedness for and response towards COVID-19.

Reviewers comment 8: 

8. Line 207, I cannot understand why the people aged between 41 and 60 years can be defined as “young adults”. They are already middle age. Also, what is the meaning of “young” here? Do the authors want to say youth?

Author response: Omitted, kept only Adult

We tried to chronologically classify as 

Adolescents 

Youth 

Young Adult (converted to Adult) 

Elderly

Reviewers comment 9: 

9. The authors reported the COVID 19 Symptoms and Co-morbidity. However, the authors did not mention how they assessed these symptoms and co-morbidity.

Author response: Discussed in the Methodology section in Questionnaire headings, Page 4-5, line (130-133) as 

The co-morbidities and symptoms of COVID-19were obtained from WHO resources and asked to the respondents’ whether present or not https://www.who.int/docs/default-source/coronaviruse/situation-reports/20200309-sitrep-49-covid-19.pdf?sfvrsn=70dabe61_4

Explanation

The symptom list is taken from the mentioned document page 9, how to suspect a case

Co-morbidities was taken from https://erj.ersjournals.com/content/early/2020/03/17/13993003.00547-2020

Reviewers comment 10: 

10. The authors mentioned that they used chi-square test in the Data analysis section; however, there are no results on chi-square values in the Results section.

Author response: Agreed, added in Table 3

Reviewers comment 11: 

11. The presentation of Table 2 is confusing. What are the “others” in the variables column? Also, the authors should explicitly indicate what the reference group is.

Author response: Table 2 statistics has been changed and defined the others

Reviewers comment 12: 

12. The authors mentioned that they performed binary logistic regression in the Data analysis section; however, I see no results on the logistic regression in the Results section.

Author response: Agreed, the analysis reformed and tested as binary logistic regression with the concern of statistician. Putted Chi-square result, Co-efficient result, Odd ratio value, and p-value

Reviewers comment 13: 

13. The meanings of SSC and HSC in the Table 3 are unclear.

Author response:Agreed, put in the table and narrated bellow as secondary school certificate and higher secondary school certificate

Reviewers comment 14: 

14. The Discussion reports redundant information; that is, the author repeated their results in the Discussion again. However, I did not see the authors “explain” their findings in the Discussion. They simply repeated the Results and then mentioned other similar studies’ findings.

Author response: Agreed and revised in each finding. 

Page 15, line 338-343

Page 15 line 354-358

Page 16 line 372-374

Page 16 line 378-384 

Also, it is difficult to explain why the study findings are such, because this is a cross-sectional study and focused on observation and relationship. Any comment on why needs in-depth findings that are missing in this field. Hence we tried best to explain.

Reviewers comment 15: 

15. There are some awkward sentences hinder the reading. For example, “Limitations response rates, correlation with fear and psychological issues, and completion of the questionnaire for COVID-19 positive. We recommend future studies to include information on limitation of movement, isolation and insomnia as it relates to psychological illness, as well as information regarding neurological signs and symptoms of patients and relationship to cognition and fear.”

Author response: We have prepared the manuscript with native English speakers; hence we agree the language can be better. We supplied the manuscript to PLOS recommended authorship services to make fit and appropriate with your comments. The language editing has been performed by professional scientific editing service AJE recommended by PLOS. 

We have revised the paragraph as- 

Limitations included the response rate (62.85%) and the completion of the questionnaire by none COVID-19-positive individuals. We recommend that future studies include information on the long-term observation of corvid-19-positive cases or cases with symptoms with respect to movement, function, physical signs, mental health, and quality-of-life issues.

Reviewers comment 16: 

16. The authors mentioned the age range of 13-90 in the Abstract. However, the age range is 13-88 in the Results section.

Author response:Apology, it was unintentional. Our eligibility criteria were 13-90 years but we found 13- 88 years. Corrected in the abstract.

Additional requirements

Additional requirements 1 2

1. Please ensure that your manuscript meets PLOS ONE's style requirements, including those for file naming

Author response: To ensure the paper fits best with the language and journal format, we sent the paper in your recommended professional scientific editing service. The language editing has been performed by professional scientific editing service AJE recommended by PLOS.

Additional requirement 3

3. Please include additional information regarding the survey or questionnaire used in the study and ensure that you have provided sufficient details that others could replicate the analyses. For instance, if you developed a questionnaire as part of this study and it is not under a copyright more restrictive than CC-BY, please include a copy, in both the original language and English, as Supporting Information. Moreover, please include more details on how the questionnaire was pre-tested, and whether it was validated. 

Author response: Supplied as additional information- questionnaire 

The supplied files are 

S1 Table.Table 1.Relationship of demographic characteristics with Knowledge and fear

S2 Table.Table 2.Results of Binary logistic regression on factors associated with Knowledge and fear

S3 Table.Table 3.The relationship among Attitude and practice with demographic variables

S1 File.Track changed Manuscript for professional scientific editing service. The track changing enabled by AJE professional scientific editing services recommended by PLOS one (http://learn.aje.com/plos/)

S2 File. English Questionnaire 

S3 File. Bangla Questionnaire

Additional requirement 4

4. In your Methods section, please provide additional information about the participant recruitment method and the demographic details of your participants. Please ensure you have provided sufficient details to replicate the analyses such as: a) the recruitment date range (month and year), b) a description of any inclusion/exclusion criteria that were applied to participant recruitment, c) a table of relevant demographic details, d) a statement as to whether your sample can be considered representative of a larger population, e) a description of how participants were recruited, and f) descriptions of where participants were recruited and where the research took place.

Author response: Supplied and mentioned each part (a-f) from Page 4- line 117 to page 5-line 153. 

Additional requirement 5

5. We noted in your submission details that a portion of your manuscript may have been presented or published elsewhere. The article is in the MedRxiv pre-print server. Please clarify whether this publication was peer-reviewed and formally published. If this work was previously peer-reviewed and published, in the cover letter please provide the reason that this work does not constitute dual publication and should be included in the current manuscript

Author response: It was a pre-print server and it was saying that your journal allows the pre-print server (https://www.medrxiv.org/submit-a-manuscript). If the paper is accepted and published in your journal, the crossref and doi will be automatically updated. The revised manuscript can be preprinted also. 

Our pre-print manuscript https://www.medrxiv.org/content/10.1101/2020.06.17.20133611v1

Additional requirement 6

6. Your ethics statement must appear in the Methods section of your manuscript. If your ethics statement is written in any section besides the Methods, please move it to the Methods section and delete it from any other section. Please also ensure that your ethics statement is included in your manuscript, as the ethics section of your online submission will not be published alongside your manuscript. 

Author response: We put in Method section only on page 5 line 139 to line 143

---

## [Decision Letter · Decision Letter 1]

11 Sep 2020

Knowledge, Attitudes, and Fear of COVID-19 during the Rapid Rise Period in Bangladesh

PONE-D-20-20624R1

Dear Dr. Hossain,

We’re pleased to inform you that your manuscript has been judged scientifically suitable for publication and will be formally accepted for publication once it meets all outstanding technical requirements.

Kind regards,

Amir H. Pakpour, Ph.D.

Academic Editor

PLOS ONE

Additional Editor Comments (optional):

Reviewers' comments:

Reviewer's Responses to Questions

**Comments to the Author**

1. If the authors have adequately addressed your comments raised in a previous round of review and you feel that this manuscript is now acceptable for publication, you may indicate that here to bypass the “Comments to the Author” section, enter your conflict of interest statement in the “Confidential to Editor” section, and submit your "Accept" recommendation.

Reviewer #1: All comments have been addressed

2. Is the manuscript technically sound, and do the data support the conclusions?

Reviewer #1: Yes

3. Has the statistical analysis been performed appropriately and rigorously? 

Reviewer #1: Yes

4. Have the authors made all data underlying the findings in their manuscript fully available?

Reviewer #1: Yes

5. Is the manuscript presented in an intelligible fashion and written in standard English?

Reviewer #1: Yes

6. Review Comments to the Author

Reviewer #1: The authors have satisfactorily respond to my prior comments. I have no more comments for the revised manuscript.

7. PLOS authors have the option to publish the peer review history of their article (what does this mean?). If published, this will include your full peer review and any attached files.

Reviewer #1: No

---

## [Editor Report · Acceptance letter]

16 Sep 2020

PONE-D-20-20624R1 

Knowledge, Attitudes, and Fear of COVID-19 during the Rapid Rise Period in Bangladesh 

Dear Dr. Hossain:

I'm pleased to inform you that your manuscript has been deemed suitable for publication in PLOS ONE. Congratulations! Your manuscript is now with our production department. 

Kind regards, 

on behalf of

Dr. Amir H. Pakpour 

Academic Editor

PLOS ONE